

# A GPU-based solution for fast calculation of the betweenness centrality in large weighted networks

Rui Fan[1], Ke Xu[1] and Jichang Zhao[2]

[1] State Key Laboratory of Software Development Environment, Beihang University, Beijing, PR China
[2] School of Economics and Management, Beihang University, Beijing, PR China

## ABSTRACT

Betweenness, a widely employed centrality measure in network science, is a decent proxy for investigating network loads and rankings. However, its extremely high computational cost greatly hinders its applicability in large networks. Although several parallel algorithms have been presented to reduce its calculation cost for unweighted networks, a fast solution for weighted networks, which are commonly encountered in many realistic applications, is still lacking. In this study, we develop an efficient parallel GPU-based approach to boost the calculation of the betweenness centrality (BC) for large weighted networks. We parallelize the traditional Dijkstra algorithm by selecting more than one frontier vertex each time and then inspecting the frontier vertices simultaneously. By combining the parallel SSSP algorithm with the parallel BC framework, our GPU-based betweenness algorithm achieves much better performance than its CPU counterparts. Moreover, to further improve performance, we integrate the work-efficient strategy, and to address the load-imbalance problem, we introduce a warp-centric technique, which assigns many threads rather than one to a single frontier vertex. Experiments on both realistic and synthetic networks demonstrate the efficiency of our solution, which achieves $2.9\times$ to $8.44\times$ speedups over the parallel CPU implementation. Our algorithm is open-source and free to the community; it is publicly available through https://dx.doi.org/10.6084/m9.figshare.4542405. Considering the pervasive deployment and declining price of GPUs in personal computers and servers, our solution will offer unprecedented opportunities for exploring betweenness-related problems and will motivate follow-up efforts in network science.

## INTRODUCTION

As an emerging multidisciplinary research area, network science has attracted much attention from researchers of various backgrounds, such as computer science, biology and physics, in recent decades. In these contributions, the betweenness centrality (BC) is often applied as a critical metric for measuring the significance of nodes or edges (*Ma & Sayama, 2015*; *Freeman, 1977*; *Barthélemy, 2004*; *Abedi & Gheisari, 2015*; *Goh et al., 2003*). For example, Girvan and Newman developed a community detection algorithm based on edge BC (*Girvan & Newman, 2002*), *Leydesdorff (2007)* used centrality as an indicator of the interdisciplinarity of scientific journals and *Motter & Lai (2002)* established a model

Corresponding author
Jichang Zhao, jichang@buaa.edu.cn

of cascading failures in which the load on a node is represented by its betweenness. However, the extremely high temporal and spatial complexity of the BC calculation greatly limits its applicability in large networks. Before the landmark work of *Brandes (2001)*, the complexity of the algorithm for computing the BC was $O(n^3)$ in time and $O(n^2)$ in space. *Brandes (2001)* reduced the complexity to $O(n + m)$ in space and $O(nm)$ and $O(nm + n^2 \log n)$ in time for unweighted and weighted networks, respectively, where $n$ is the number of vertices and $m$ is the number of edges. However, this improved algorithm still cannot satisfy the requirements for scientific computations in the present era of information explosion, as an increasing number of unexpectedly large networks emerge, such as online social networks, gene networks and collaboration networks. For example, Twitter has hundreds of millions of active users, who form an enormous online social network. However, calculating the BC of a weighted network with one million nodes may take approximately one year, which is an unsupportable time cost. Existing parallel CPU algorithms may reduce this time to several months; however, this is still too expensive. Because of this problem, there is a pressing need to develop faster BC algorithms for the exploration of diverse domains.

General-purpose GPU (GPGPU) computing, which has high parallelization, provides an opportunity to employ parallel algorithms implemented on GPUs to achieve better performance. For network-related problems, researchers have devoted efforts to conquering irregular graph structures using GPGPU techniques and have achieved higher performance than is possible with traditional sequential CPU algorithms (*Mitchell & Frank, 2017*; *Merrill, Garland & Grimshaw, 2015*; *Wang et al., 2015*; *Harish & Narayanan, 2007*; *Cong & Bader, 2005*). CUDA, developed by Nvidia Corporation, is the most popular GPU computing framework, and some researchers have even used this framework to parallelize the Brandes algorithm (*Shi & Zhang, 2011*; *Sariyüce et al., 2013*; *McLaughlin & Bader, 2014*, *2015*). However, previous works have concentrated on unweighted networks for simplicity, although to the best of our knowledge, many realistic networks are weighted ones. The most significant difference in the BC algorithm between unweighted and weighted networks is the shortest-path calculation. In weighted networks, the Dijkstra algorithm should be used to solve the single-source shortest path (SSSP) problem rather than the breadth-first search (BFS) algorithm. In previous work, many efforts have been devoted to developing a GPU version of the SSSP problem using the well-known Dijkstra algorithm (*Martin, Torres & Gavilanes, 2009*; *Ortega-Arranz et al., 2013*; *Delling et al., 2011*; *Davidson et al., 2014*). Although such algorithms have been successfully developed and presented, establishing a parallel version of the BC algorithm for weighted networks is nontrivial because the original SSSP algorithm must be modified in many critical aspects for this task, and to the best of our knowledge, a suitable fast solution is still lacking. With the aim of filling this vital gap, we propose a fast solution using CUDA for calculating the BC in large weighted networks based on previous GPU BC algorithms and SSSP algorithms.

To make our algorithm more efficient, we make efforts to optimize it by employing several novel techniques to overcome the influence of irregular network structures. Real-world networks have many characteristics that can degrade the performance of

GPU parallelization algorithms. For example, the frontier set of nodes is always small compared with the total number of vertices, especially for networks with large diameters. At the same time, the majority of the nodes do not need to be inspected in each step; hence, processing all vertices simultaneously, as is done in traditional algorithms, is wasteful. *McLaughlin & Bader (2014)* proposed a work-efficient strategy to overcome this problem. Another well-known issue is that the power-law degree distribution in realistic networks induces severe load imbalance. Several methods have been proposed in previous studies to overcome this problem; e.g., *Jia et al. (2011)* employed an edge parallel strategy to avoid load imbalance, and *Hong et al. (2011)* addressed this problem by using a warp technique. In this paper, we systematically investigate the advantages and disadvantages of these previous methods and incorporate them into our algorithm to solve the above two problems. Experiments on both real-world and synthetic networks demonstrate that our algorithm significantly outperforms the baseline GPU algorithm. Our main contributions are as follows:

- Based on previous GPU-based parallel SSSP and BC algorithms, we propose an efficient algorithm for calculating the BC for weighted networks, which achieves $2.9\times$ to $8.44\times$ speedups over the parallel CPU algorithm on realistic networks.
- We compare the traditional node-parallel method with the work-efficient version and the warp-centric method. Experiments on realistic networks and synthetic networks demonstrate that the combination of these two strategies performs better than either the basic node-parallel method or the individual strategies; it achieves an average speedup of $2.34\times$ over the baseline method on realistic networks.
- We package our algorithm as a useful tool that can be used to calculate both node and edge BC on weighted networks. Researchers can apply this tool to quickly and conveniently calculate BC values for weighted networks, especially large networks. The source code is publicly available through https://dx.doi.org/10.6084/m9.figshare.4542405.

## BACKGROUND

First, we briefly introduce the well-known Brandes algorithm and Dijkstra algorithm based on preliminary definitions of a network and the BC.

### Brandes algorithm

A graph can be denoted by $G(V, E)$, where $V$ is the set of vertices and $E$ is the set of edges. An edge can be denoted by $(u, v, w)$, which means that there is a link of weight $w$ connecting nodes $u$ and $v$. If edge $(u, v)$ exists, it can be traversed either from $u$ to $v$ or from $v$ to $u$ because we focus only on undirected graphs in this paper. For directed graphs, if only an edge $(u, v)$ exists, then the algorithm will store only the edge $(u, v)$ and will process only this edge when inspecting vertex $u$; it will ignore $(v, u)$ when inspecting vertex $v$. Thus, our algorithm can easily be extended to directed graphs. A path $P = (s, \ldots, t)$ is defined as a sequence of vertices connected by edges, where $s$ is the starting node and $t$ is the ending node. The length of $P$ is the sum of the weights of the edges contained in $P$. $d(s, t)$ is the distance between $s$ and $t$, which is the length of the shortest path

connecting $s$ and $t$. $\sigma_{st}$ denotes the number of shortest paths from $s$ to $t$. In accordance with these definitions, we have $d(s, s) = 0$, $\sigma_{ss} = 1$, $d(s, t) = d(t, s)$ and $\sigma_{st} = \sigma_{ts}$ for an undirected graph. $\sigma_{st}(v)$ denotes the number of shortest paths from $s$ to $t$ that include $v$. Based on these definitions, the BC can be defined as:

$$C_B(v) = \sum_{s \neq v \neq t \in V} \frac{\sigma_{st}(v)}{\sigma_{st}}. \tag{1}$$

From the above definitions, the calculation of the BC can be naturally separated into the following two steps:

1. Compute $d(s, t)$ and $\sigma_{st}$ for all node pairs $(s, t)$.
2. Sum all pair dependencies.

Here, a pair dependency is defined as $\delta_{st}(v) = \frac{\sigma_{st}(v)}{\sigma_{st}}$. The time complexity of the first step is $O(mn)$ or $O(mn + n^2\log n)$ for an unweighted graph or a weighted graph, respectively; therefore, the bottleneck of this algorithm is the second step, which has a time complexity of $O(n^3)$. Brandes developed a more efficient BC algorithm with a time complexity of $O(mn)$ for unweighted graphs and $O(mn + n^2\log n)$ for weighted graphs. The critical point is that the dependency of a node $v$ for a source node $s$ is $\delta_s(v) = \sum_{u:v \in P_s(u)} \frac{\sigma_{sv}}{\sigma_{su}} (1 + \delta_s(u))$. By applying this equation, we can accumulate the dependencies after computing the distances and numbers of shortest paths only from the source vertex $s$ to all other vertices, rather than after computing the shortest paths for all pairs.

We can easily develop a parallel version of the Brandes algorithm for unweighted graphs because the graph is always traversed as a tree using the BFS algorithm. Given a source node $s$, the root of the tree is $s$, and the tree is produced using the BFS method in the first step. In the second step, dependencies related to the source node $s$ are calculated from the leaves to the root of the tree, and nodes at the same level are isolated and have no influence on each other. As a result, the parallel version of the algorithm can simultaneously explore all nodes at the same level in both steps, thereby fundamentally accelerating the BC calculation.

## Dijkstra algorithm

The Dijkstra algorithm (*Dijkstra, 1959*) and the Floyd–Warshall algorithm (*Floyd, 1962*) are commonly employed to solve shortest-path problems. The Dijkstra algorithm is more easily adaptable to the BC problem because the Brandes algorithm accumulates dependencies after computing SSSPs rather than after finding and storing the shortest paths for all pairs. The Dijkstra algorithm applies a greedy strategy to solve the SSSP problem. In this algorithm, the source node is $s$, and once the shortest path from $s$ to another node $u$ is found, $u$ will be settled. As seen in Algorithm 1, all nodes in graph $G$ are separated into two sets: the settled vertices and the unsettled vertices $Q$. An array $D$ is used to store tentative distances from $s$ to all nodes. Initially, $Q$ stores all nodes, $D(s) = 0$, and $D(u) = \infty$ for all other nodes (lines 1–5). During iteration (lines 6–9), the node $u$ with

| **Algorithm 1** Sequential Dijkstra Algorithm. |
|---|
| 1: $Q \leftarrow$ empty set |
| 2: **for** $v \in V$ **do** |
| 3:     $D[v] \leftarrow \infty$ |
| 4:     add $v$ to $Q$ |
| 5: $D[s] \leftarrow 0$ |
| 6: **while** $Q$ is not empty **do** |
| 7:     $u \leftarrow Extract\_Min(Q)$ |
| 8:     **for** $v \in neighbors(u)$ **do** |
| 9:         $D[v] \leftarrow D[u] + weight_{uv}$ |

the shortest tentative distance $D[u]$ (denoted by $Extract\_Min(Q)$) is selected and settled, which means that the shortest path to node $u$ is found and $D[u]$ is set to the corresponding value. Then, for each node $v \in neighbors(u)$, if $D[u] + w(u, v) < D[v]$, $D[v]$ will be updated to $D[u] + w(u, v)$. The above procedures, in which one node is settled and the tentative distances of its neighbors are then updated, are repeated until $Q$ is empty, i.e., until all nodes in graph $G$ have been settled. According to the above description, the Dijkstra algorithm has no parallel characteristics because it selects one frontier node in each iteration. However, this restriction can be loosened to allow several frontier vertices to be explored simultaneously, in a manner similar to the BFS parallel approach.

## RELATED WORK

### Graph traversal strategies

For unweighted networks, the Brandes algorithm applies the traditional BFS strategy in the shortest-path step. The BFS algorithm produces a traversal tree, which can later be used in the dependency accumulation step. This behavior makes it easy to parallelize both steps of the unweighted BC algorithm; i.e., threads are assigned to all vertices in the graph, and if a vertex is in the frontier set, the relevant thread traverses all edges connected to that vertex. *Jia et al. (2011)* implemented their BC algorithm based on this node-parallel traversal strategy. However, this simple strategy induces a problem of load imbalance since the degrees of different vertices varies, especially in scale-free networks. Threads processing low-degree vertices must wait for threads processing high-degree vertices, which significantly slows down the calculation. Instead of assigning threads to all vertices, *Jia et al. (2011)* proposed an edge-parallel strategy in which threads are assigned to all edges and edges that are connected to frontier vertices are then inspected. This technique eliminates the load-imbalance problem. Jia et al. applied both coarse-grained and fine-grained parallelism. In a modern GPU, the kernel can employ multiple blocks, and each block contains multiple threads. In their program, the GPU employs multiple blocks, each of which focuses on one root vertex $s$ (coarse-grained parallelism). The threads within each block work cooperatively to traverse the edges in both the SSSP step and the dependency accumulation step (fine-grained parallelism). As a result, in each block,

the dependencies of all vertices related to $s$ are covered. For a vertex $w$, the dependency for the root vertex $s$ is denoted by $\delta_s[w]$. Based on the Brandes algorithm, the betweenness of $w$ can be calculated as $\sum_{s \neq w \in V} \delta_s[w]$.

## Work-efficient technique

In the edge- and node-parallel strategies, threads are assigned to all vertices or edges, respectively, and the algorithm then checks whether the vertices and edges need to be inspected; this incurs a considerable unnecessary cost because the frontier nodes might be small in size, especially for graphs of large diameters. To address this problem, *McLaughlin & Bader (2014*, *2015*) proposed an excellent work-efficient technique. In this method, a queue $Q_{curr}$ that stores the frontier vertices is maintained, and threads are assigned only to vertices that are in $Q_{curr}$. In the BFS procedure, new frontier nodes are added to $Q_{next}$. After the BFS step, the vertices in $Q_{next}$ are transferred to $Q_{curr}$, and $Q_{next}$ becomes an empty queue. To implement this technique, it is necessary to know the lengths of both queues ($Q_{curr\_len}$ and $Q_{next\_len}$) because $Q_{curr}$ and $Q_{next}$ are implemented using arrays in the GPU kernel code. For the parallel BC algorithm proposed in this paper, we develop a work-efficient version of the algorithm based on this idea.

## The issue of load-imbalance

The work-efficient algorithm still suffers from the load-imbalance problem since it is based on the node-parallel strategy. In addition to the edge-parallel strategy, other techniques have also been developed to solve this problem (*Davidson et al., 2014*; *Hong et al., 2011*). Hong et al. proposed the warp-centric concept, in which a warp rather than a thread is allocated to each node. In the modern CUDA framework, a warp consists of 32 threads, which act as a single instruction multiple data (SIMD) unit. Because a group of threads is assigned to a single frontier vertex, each thread processes a subset of the edges connected to that vertex. As a result, each thread does less work for high-degree nodes, thereby greatly reducing the waiting time. Other techniques for addressing the load-imbalance problem include Cooperative Blocks, CTA + Warp + Scan and load-balanced partitioning (*Davidson et al., 2014*; *Wang et al., 2016*). These methods attempt to assign threads to edges that need to be inspected by means of the design of several novel data structures and algorithms, which ensure excellent within-block and interblock load balance. However, these techniques require blocks to work cooperatively; i.e., each block must process several vertices or edges. Our BC algorithm applies both coarse-grained and fine-grained approaches. For this reason, we apply the warp-centric technique in our algorithm to address the load-imbalance problem.

## Parallel SSSP algorithm

To compute the BC on a weighted network, a parallel SSSP algorithm is applied in the shortest-path step. The Dijkstra algorithm, the traditional method of solving this problem, is inherently sequential because it selects a single frontier vertex in each iteration. The Bellman–Ford algorithm is easier to parallelize, but it suffers from rather low efficiency compared with the Dijkstra algorithm. *Martin, Torres & Gavilanes (2009)*

proposed a parallel Dijkstra algorithm, in which all vertices with the minimum tentative distance are inserted into the frontier set and the vertices in the frontier set are then processed simultaneously. *Ortega-Arranz et al. (2013)* implemented a more aggressive algorithm. They loosened the condition for selecting frontier nodes, resulting in more than one frontier node in each iteration, to achieve higher parallelism. δ-Stepping is another frequently employed parallel SSSP algorithm, in which vertices are grouped into buckets and all vertices in a bucket are processed simultaneously. However, as described in *Davidson et al. (2014)*, there are three main characteristics that make δ-stepping difficult to implement efficiently on a GPU; e.g., it requires dynamic arrays, which are poorly supported in the CUDA framework. Because of this, we base our SSSP algorithm on the parallel Dijkstra algorithm. However, previous SSSP algorithms have focused only on the values of the shortest paths, neglecting the number of shortest paths, which is also necessary for the BC calculation. In this paper, we modify the parallel Dijkstra algorithm presented in *Ortega-Arranz et al. (2013)* to combine it smoothly with our BC algorithm for weighted graphs.

# GPU-BASED ALGORITHM

Our GPU-based BC algorithm for weighted graphs applies both coarse-grained (in which one block processes one root vertex $s$) and fine-grained (in which all threads in a block compute the shortest paths and dependencies related to $s$) parallel strategies. In a block, the shortest paths and dependencies corresponding to the root vertex processed by that block are calculated using Brandes's two-step framework. In the shortest-path step, we build a multi-level structure from root to leaves by relaxing the condition that a single frontier node is selected in each iteration and then calculating the distances and numbers of shortest paths for all selected frontier nodes simultaneously. In the dependency accumulation step, the multi-level structure built in the first step is re-employed to calculate the dependencies of the vertices from the leaves to the root of the multi-level structure. Calculations for vertices at the same level are performed simultaneously.

## Parallel BC algorithm

In this section, we introduce the details of our GPU version of the BC algorithm for weighted graphs. First, we apply the *compressed sparse row* (CSR) format, which is widely used in graph algorithms, to store the input graph (*Bell & Garland, 2009*; *Davidson et al., 2014*). This format is space efficient because both a vertex and an edge consume one entry, and it is convenient for performing the traversal task on a GPU. Moreover, edges related to the same vertex are stored consecutively in memory, which makes the warp-centric technique more efficient. For the storage of weighted graphs, an additional array is required to store the weights of all edges.

We apply both coarse-grained and fine-grained parallel strategies. The pseudo-code presented in this paper describes the parallel procedure for threads within a block. Algorithm 2 shows the initialization of the required variables. $U$ and $F$ represent the unsettled set and the frontier set, respectively. $v$ is unsettled if $U[v] = 1$ and is a frontier

---

**Algorithm 2** BC: variable initialization.

1:  **for** $v \in V$ **do in parallel**
2:      $U[v] \leftarrow 1$
3:      $F[v] \leftarrow 0$
4:      $d[v] \leftarrow \infty$
5:      $\sigma[v] \leftarrow 0$
6:      $\delta[v] \leftarrow 0$
7:      $lock[v] \leftarrow 0$
8:      $ends[v] \leftarrow 0$
9:      $S[v] \leftarrow 0$
10: $d[s] \leftarrow 0$
11: $\sigma[s] \leftarrow 1$
12: $U[s] \leftarrow 0$
13: $F[s] \leftarrow 1$
14: $S[0] \leftarrow s; S_{len} \leftarrow 1$
15: $ends[0] \leftarrow 0; ends[1] \leftarrow 1; ends_{len} \leftarrow 2$
16: $\Delta \leftarrow 0$

---

node if $F[v] = 1$. $d$ represents the tentative distance, and $\sigma[v]$ is the number of shortest paths from $s$ to $v$. $\delta[v]$ stores the dependencies of $v$. $lock$ stores locks for all nodes to avoid race conditions. If $lock[v] = 1$, changing neither $\sigma[v]$ nor $\delta[v]$ is permitted (see the next section for details). Vertices at the same level are consecutively recorded in $S$, and the start (or end) point of each level in $S$ is stored in $ends$. In other words, $S$ and $ends$ record the levels of traversal in the CSR format; they are used in the dependency accumulation step. As seen in Algorithm 3, in the dependency accumulation step, we obtain all nodes at the same level from $S$ and $ends$ and accumulate the dependencies of these nodes simultaneously. Note that in Algorithm 3, we assign threads only to nodes that need to be inspected rather than to all nodes, which enhances the efficiency of the algorithm by avoiding redundant threads. We update the dependencies of edges in line 12 of Algorithm 3 if edge betweenness is required.

## Parallel Dijkstra algorithm

The parallel version of the BFS procedure that is applied in the BC algorithm for unweighted networks can be naturally adapted from the sequential version because vertices located on the same level in the BFS tree can be inspected simultaneously. Moreover, in the dependency accumulation step (step two), dependencies are calculated from low-level vertices (nodes with the greatest depths in the tree) to high-level vertices (nodes that are close to the source node), and calculations for nodes at the same level are again performed simultaneously. In the weighted version, a multi-level structure is similarly necessary in the dependency accumulation step to achieve parallelization. As seen in Fig. 1A, this structure should satisfy the condition $\forall\, u \in P_v,\, l_u < l_v$, where $l_i$ denotes

---

**Algorithm 3** BC: Dependency Accumulation.

1: $depth \leftarrow ends_{len} - 1$

2: **while** $depth > 0$ **do**

3:     $start \leftarrow ends[depth - 1]$

4:     $end \leftarrow ends[depth] - 1$

5:     **for** $0 \leq i \leq end - start$ **do in parallel**

6:         $w \leftarrow S[start + i]$

7:         $dsw \leftarrow 0$

8:         **for** $v \in neighbors(w)$ **do**

9:             **if** $d[v] = d[w] + weight_{wv}$ **then**

10:                 $c \leftarrow \sigma[w]/\sigma[v] * (1 + \delta[v])$

11:                 $dsw \leftarrow dsw + c$

12:                 $atomicAdd(edgeBC[w], c)$

13:         $\delta[w] \leftarrow dsw$

14:         **if** $w \neq s$ **then**

15:             $atomicAdd(BC[w], \delta[w])$

16:     $depth \leftarrow depth - 1$

---

the level of node $i$ in the multi-level structure and $P_i$ represents the set of predecessors of vertex $i$. Previous high-performance parallel SSSP algorithms have calculated only the shortest-path values, neglecting the number of shortest paths and the level relationships. In this paper, we propose a variant of the parallel Dijkstra algorithm that produces both the number of shortest paths and the multi-level structure needed in our betweenness algorithm.

In the sequential Dijkstra algorithm, the fact that one frontier node is selected in each iteration makes parallelization a difficult task. However, this restriction can be relaxed, which means that several nodes can be settled at once to form the frontier set, allowing them to be inspected simultaneously in the next step. Moreover, these settled nodes satisfy the level condition, and because of this, they form a new level to be inspected simultaneously in the dependency accumulation step. In this paper, we apply the method described in *Ortega-Arranz et al. (2013)*. In this method, $\Delta_{node\ v} = \min(w(v, u): (v, u) \in E)$ is precomputed. Then, we define $\Delta_i$ as

$$\Delta_i = \min\{(D(u) + \Delta_{node\ u}) : u \in U_i\}, \tag{2}$$

where $D(u)$ is the tentative distance of node $u$ and $U_i$ is the set of unsettled nodes in iteration $i$. All nodes that satisfy the condition

$$D(v) \leq \Delta_i \tag{3}$$

are settled and become frontier nodes. When the Dijkstra algorithm is applied in the BC calculation, the number of shortest paths should be counted, and predecessor relationships between vertices at the same level are not permitted; otherwise, the parallel

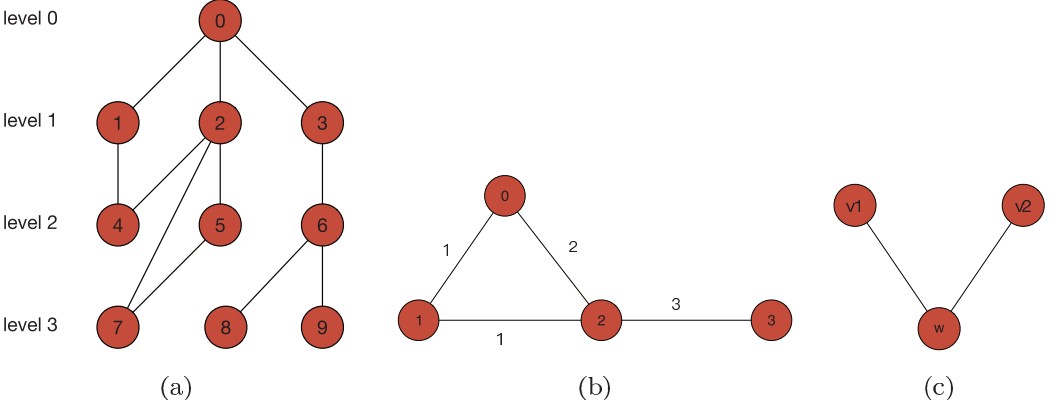

**Figure 1** (A) **An example of a multi-level structure.** It is built in the SSSP step and will later be used in the dependency accumulation step. Nodes at the same level are inspected simultaneously in both steps. (B) An example of the selection of a set of frontier nodes in which using Eq. (3) will cause the number of shortest paths calculated for $v_3$ to be incorrect. (C) An example of a race condition. $v_1$ and $v_2$ are both frontier nodes in the same iteration, and both are connected to $w$.

algorithm would result in incorrect dependencies. To this end, the above condition should be modified to

$$D(v) < \Delta_i. \tag{4}$$

Figure 1B illustrates an example, in which the vertex $v_0$ is the source node. If Eq. (3) were to be applied, $v_1$ and $v_2$ would become the frontier nodes after the inspection of $v_0$ in the first iteration, and the number of shortest paths would be 1 for both $v_1$ and $v_2$. Then, $v_1$ and $v_2$ would be inspected simultaneously in the next step. If $v_2$ were to be processed first, the number of shortest paths for $v_3$ would be set to 1; however, the correct number of shortest paths for $v_3$ is 2. This mistake arises from the overambitious condition defined in Eq. (3); $v_2$ should not be settled after the first iteration. Although the distances for all nodes would be correct with Eq. (3), the numbers of shortest paths would be wrong. By contrast, Eq. (4) will lead to the correct number of shortest paths for $v_3$ because only $v_1$ will be settled after the first iteration. This condition appears on line 29 in Algorithm 4.

By applying Eq. (4) in the SSSP step, we achieve the correct numbers of shortest paths and construct a multi-level structure by setting each set of frontier nodes as a new level.

Algorithm 4 presents our parallel Dijkstra algorithm in detail. The tentative distance and number of shortest paths are calculated as shown in lines 2–13. For a frontier vertex $v$, the thread inspects all edges connected to $v$. For an edge $(v, w)$, if it finds a shorter path from $v$, i.e., $d[v] + weight_{vw} < d[w]$, $d[w]$ will be updated, and $\sigma[w]$ will be set to zero since the previous number of shortest paths is invalid. Then, if $d[w] = d[v] + weight_{vw}$, the number of shortest paths for vertex $w$ will be updated to $\sigma[w] + \sigma[v]$ in accordance with the Brandes algorithm. In this way, both the value and number of shortest paths are calculated and stored. In this part of the calculation, a race condition problem may arise because multiple nodes in the frontier set may connect to the same node, as seen in

| Algorithm 4  BC: shortest-path calculation using the Dijkstra algorithm. |
| :--- |

1:  **while** $\Delta < \infty$ **do**

2:      **for** $v \in V$ and $F[v] = 1$ **do in parallel**

3:          **for** $w \in neighbors(v)$ **do**

4:              $needlock \leftarrow true$

5:              **while** $needlock$ **do**

6:                  **if** $0 = atomicCAS(lock[w],0,1)$ **then**

7:                      **if** $U[w] = 1$ **and** $d[v] + weight_{vw} < d[w]$ **then**

8:                          $d[w] \leftarrow d[v] + weight_{vw}$

9:                          $\sigma[w] \leftarrow 0$

10:                     **if** $d[w] = d[v] + weight_{vw}$ **then**

11:                         $\sigma[w] \leftarrow \sigma[w] + \sigma[v]$

12:                     $atomicExch(lock + w, 0)$

13:                     $needlock \leftarrow false$

14:     $\Delta \leftarrow \infty$

15:     **for** $v \in V$ **do in parallel**

16:         **if** $U[v] = 1$ **and** $d[v] < \infty$ **then**

17:             $atomicMin(\Delta, d[v] + \Delta_{node\ v})$

18:     $cnt \leftarrow 0$

19:     **for** $v \in V$ **do in parallel**

20:         $F[v] \leftarrow 0$

21:         **if** $U[v] = 1$ **and** $d[v] < \Delta$ **then**

22:             $U[v] \leftarrow 0$

23:             $F[v] \leftarrow 1$

24:             $t \leftarrow atomicAdd(S_{len}, 1)$

25:             $S[t] \leftarrow v$

26:             $atomicAdd(cnt, 1)$

27:     **if** $cnt > 0$ **then**

28:         $ends[ends_{len}] \leftarrow ends[ends_{len} - 1] + cnt$

29:         $ends_{len} \leftarrow ends_{len} + 1$

**Fig. 1C.** In this example, both $v_1$ and $v_2$ are in the frontier set and are connected to $w$, which results in the classical race condition problem. The reason this is a problem is that two or more threads may attempt to modify $d[w]$ or $\sigma[w]$ simultaneously. To avoid this, we define a lock for each node. The first thread to focus on $w$ will be granted the lock, and other threads will not be permitted to change $d[w]$ and $\sigma[w]$. We also adopt an atomic operation *atomicCAS* and a variable *needlock*. For all threads, *needlock* is initially true (line 4), and the threads will enter the following iteration. If a thread is granted the lock for $w$, it will run the shortest-path procedure and then release the lock (line 12), and *needlock = false* will be assigned (line 13) to exit the loop. If another thread owns

---

**Algorithm 5** Work-efficient BC: variable initialization.

1: **for** $v \in V$ **do in parallel**

2:     // initialize all variables except $F$

3: $F[0] \leftarrow s$

4: $F_{len} = 1$

5: // initialize other variables

---

**Algorithm 6** Work-efficient BC: shortest-path calculation using the Dijkstra algorithm.

1: **while** $\Delta < \infty$ **do**

2:     **for** $0 \le i < F_{len}$ **do in parallel**

3:         $v \leftarrow F[i]$

4:         // inspect $v$

5:         // calculate $\Delta$

6:     $F_{len} \leftarrow 0$

7:     **for** $v \in V$ **do in parallel**

8:         **if** $U[v] = 1$ **and** $d[v] < \Delta$ **then**

9:             $U[v] \leftarrow 0$

10:             $t \leftarrow atomicAdd(F_{len}, 1)$

11:             $F[t] \leftarrow v$

12:     **if** $F_{len} > 0$ **then**

13:         $ends[end\,s_{len}] \leftarrow end\,s[end\,s_{len} - 1] + F_{len}$

14:         $end\,s_{len} \leftarrow end\,s_{len} + 1$

15:         **for** $0 \le i < F_{len}$ **do in parallel**

16:             $S[S_{len} + i] \leftarrow F[i]$

17:         $S_{len} \leftarrow S_{len} + F_{len}$

---

the lock, the thread will run the circulation but do nothing until the other thread releases the lock. In this way, all threads that need to inspect vertex $w$ can perform the shortest-path task while avoiding race conditions. The lock cannot be replaced with an atomic operation because in the shortest-path procedure, multiple instructions related to $w$ (from lines 7 to 13) are executed, rather than only one, and they cannot be interrupted by other threads that may modify $d[w]$ and $\sigma[w]$. After $d$ and $\sigma$ have been computed for all nodes, we can obtain $\delta_i$ based on the computed results, as seen on lines 14–17. Finally, $U$, $F$, $S$ and $ends$ are updated for the next iteration.

## Work-efficient method

As seen on line 2 in Algorithm 4, threads will be assigned to all nodes, but calculations will be performed only for nodes in the frontier set, which may be inefficient. *McLaughlin & Bader (2014, 2015)* developed an excellent work-efficient technique for solving this problem. In this paper, we develop a work-efficient version of our algorithm by adopting

---

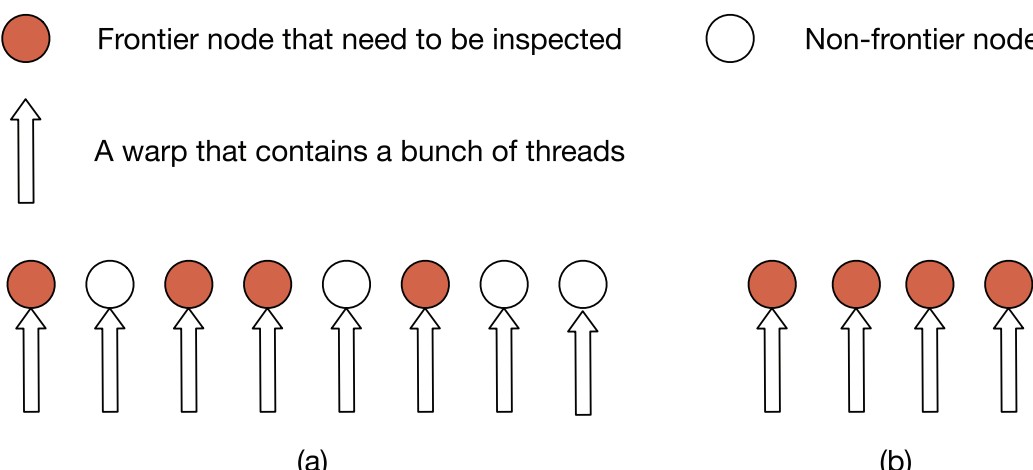

**Figure 2 Examples of thread allocation using (A) the node-parallel method and (B) the work-efficient method.** Red nodes and white nodes are frontier and non-frontier nodes, respectively, and each arrow represents a warp, which contains multiple threads that are all assigned to the same node. In the warp-centric method, more threads will be wasted on non-frontier nodes that do not need to be inspected. However, this problem can be solved by combining the warp-centric and work-efficient methods, as shown in (B).           

this idea. $F$ is replaced with a *queue* that stores all frontier nodes, and a variable $F_{len}$ is defined to recode the length of $F$, as seen in Algorithm 5. Then, at line 2 in Algorithm 6, threads can be assigned to $F[0] \sim F[F_{len} - 1]$, which may be much smaller than the total number of nodes. At the same time, the method used to update $F$ should also be modified as shown in Algorithm 6.

## Warp-centric method

Many real-world networks are scale-free in nature, which means that their degree distributions follow a power law. When parallel graph algorithms are implemented using the node-parallel strategy, this feature gives rise to a severe load-imbalance problem. Most nodes have low degrees, while some nodes have extremely high degrees. Threads that are assigned to high-degree nodes will run slowly, and other threads will have to wait. The edge-parallel strategy can be used to solve this problem (*Jia et al., 2011*), but it simultaneously introduces other problems of underutilization. In this paper, we apply the novel warp-centric method (*Hong et al., 2011*), in which a warp rather than a thread is allocated to a single node. Then, each thread within a warp focuses on a subset of the edges connected to the corresponding node. As a result, each thread does less work for nodes with high degrees, and the wait time will be greatly decreased. Moreover, memory access patterns can be more tightly grouped compared with conventional thread-level task allocation, and consequently, the efficiency of memory access can also be fundamentally improved.

Nevertheless, the warp-centric method also has some disadvantages. First, the degree of a node may be smaller than the warp size, which is always 32 in modern GPUs. To solve this problem, *Hong et al. (2011)* proposed virtual warps. Second, the number of required threads will be increased overall because each node needs *WARP_SIZE* threads

rather than one thread in this approach (here, *WARP_SIZE* denotes the number of threads in each warp). However, the number of threads per block is fixed; hence, each thread will be iteratively assigned to additional nodes, which may result in low performance. We find that the work-efficient technique can effectively relieve this problem because it requires fewer threads compared with the conventional node-parallel method, as seen in Fig. 2. In this paper, we apply the warp-centric method in combination with both the node-parallel and work-efficient methods, resulting in four algorithms with different thread allocation strategies, and we compare these algorithms on both real-world and synthetic networks.

## EXPERIMENTS

### Networks and settings

We collected ten weighted real-world networks from the Internet. These networks are of a broad variety of types, including collaboration networks, biological networks and social networks. We also downloaded a large synthetic network with $2^{20}$ vertices and more than 44 million edges. These networks are publicly available on the Internet and have been analyzed extensively in previous studies (*Rossi & Ahmed, 2015*; *Bansal et al., 2007*; *Palla et al., 2008*; *Barabási & Albert, 1999*; *Leskovec & Krevl, 2014*; *De Domenico et al., 2013*; *Leskovec, Adamic & Huberman, 2007*; *Bader et al., 2012*, *2014*). The details of these networks are listed in Table 1. We developed a parallel CPU algorithm based on graph-tool (https://graph-tool.skewed.de), which is an efficient network analysis tool whose core data and algorithms are implemented in C++, making it efficient for various graph-related algorithms, including betweenness calculations (https://graph-tool.skewed.de/performance). The BC calculation performed by this tool relies on the Boost Graph Library (*Siek, Lee & Lumsdaine, 2001*), and it supports the execution of parallel betweenness algorithms on weighted networks (*Gregor & Lumsdaine, 2005*) (http://www.boost.org/doc/libs/1_65_1/libs/graph_parallel/doc/html/index.html). We ran our four GPU implementations on a GeForce GTX 1080 using the CUDA 8.0 Toolkit. The GeForce GTX 1080 is a compute-capable 6.1 GPU designed using the Pascal architecture that has 20 multiprocessors, 8 GB of device memory, and a clock frequency of 1,772 MHz. The CPU we used is an Intel Core i7-7700K processor. The Core i7-7700K has a frequency of 4.2 GHz, an 8 MB cache and eight physical processor cores. We used four threads since hyperthreading does not improve performance, and we also ran a sequential version because such implementations are still widely applied by network researchers.

To further investigate the effects of network structures on the algorithms' performance, we generated two types of networks: Erdös–Rényi (ER) random graphs (*Erdös & Rényi, 1959*) and Kronecker graphs (*Leskovec et al., 2010*). The degree distribution of an ER random graph is a Poisson distribution, meaning that its node degrees are relatively balanced. Meanwhile, a Kronecker graph possesses scale-free and small-world characteristics, making it appropriate for studying the load-imbalance problem. We uniformly assigned random edge weights ranging from 1 to 10, as done in previous studies (*Martin, Torres & Gavilanes, 2009*; *Ortega-Arranz et al., 2013*). We used these synthetic networks to study the relationship between the graph structure and the traversal strategy.

**Table 1** Details of networks from public datasets.

| Network | Vertices | Edges | Max degree | Average degree | Description |
|---|---|---|---|---|---|
| bio-human-gene1 (*Rossi & Ahmed, 2015*; *Bansal et al., 2007*) | 22,283 | 12,345,963 | 7,940 | 1,108.11 | Human gene regulatory network |
| bio-human-gene2 (*Rossi & Ahmed, 2015*; *Bansal et al., 2007*) | 14,340 | 9,041,364 | 7,230 | 1,261.00 | Human gene regulatory network |
| bio-mouse-gene (*Rossi & Ahmed, 2015*; *Bansal et al., 2007*) | 45,101 | 14,506,196 | 8,033 | 643.28 | Mouse gene regulatory network |
| ca-MathSciNet-dir (*Rossi & Ahmed, 2015*; *Palla et al., 2008*) | 391,529 | 873,775 | 496 | 4.46 | Co-authorship network |
| actors (*Barabási & Albert, 1999*) | 382,219 | 15,038,094 | 3,956 | 78.69 | Actor collaboration network |
| rt-higgs (*Leskovec & Krevl, 2014*; *De Domenico et al., 2013*) | 425,008 | 732,827 | 31,558 | 3.45 | Twitter retweeting network |
| mt-higgs (*Leskovec & Krevl, 2014*; *De Domenico et al., 2013*) | 116,408 | 145,774 | 11,957 | 2.50 | Twitter mention network |
| rec-amazon (*Rossi & Ahmed, 2015*; *Leskovec, Adamic & Huberman, 2007*) | 91,813 | 125,704 | 5 | 2.74 | Product copurchase network |
| sc-shipsec1 (*Rossi & Ahmed, 2015*; *Bader et al., 2012*) | 140,385 | 1,707,759 | 67 | 24.33 | Scientific computing network |
| soc-pokec (*Leskovec & Krevl, 2014*) | 1,632,803 | 30,622,564 | 14,854 | 27.32 | Pokec social network |
| kron_g500 (*Bader et al., 2014*) | 1,048,576 | 44,619,402 | 131,503 | 112.22 | Large Kronecker network |

## Results

### Overall performance

From Table 2, we can see that all of the GPU programs achieve better performance than both the sequential and parallel CPU versions on the real-world networks. The best GPU algorithm for each network achieves speedups of 2.9× to 8.44× compared with the parallel CPU method and of 10× to 20× compared with the sequential CPU algorithm, and the performance can be markedly improved by assigning an appropriate *WARP_SIZE*. Even on the two large networks with more than one million vertices, our algorithm can produce results within 2 or 3 days. Note that the performance of previous GPU-based BC algorithms for unweighted networks might be superior to ours on networks of similar sizes because the complexity of the weighted BC algorithm is higher than that of its unweighted counterparts.

As seen from Table 2, the work-efficient method is more efficient than the node-parallel method on all networks, whereas the warp-centric method performs better on high-degree networks, such as the three biological networks. However, combining the warp-centric method and the work-efficient method always results in superior or approximately equal performance compared with the work-efficient method alone because it causes fewer threads to be required in each step, which, in turn, alleviates the second disadvantage of the warp-centric method. For networks with low average degrees, such as ca-MathSciNet-dir, rt-higgs and mt-higgs, applying the warp-centric method with the real *WARP_SIZE* (32) is always inefficient because the nodes' degrees are always smaller than *WARP_SIZE*. Using a smaller virtual *WARP_SIZE* enables better performance on

**Table 2 Benchmark results of various BC algorithms on weighted graphs, including a sequential CPU algorithm, a four-thread CPU algorithm, and node-parallel (NP), work-efficient (WE) and warp-centric (warpx denotes that the _WARP_SIZE_ is x) algorithms.**

| Algorithm | bio-human-gene1 | bio-human-gene2 | bio-mouse-gene | ca-MathSciNet-dir | actors |
|---|---|---|---|---|---|
| CPU (sequential) | 7,494.09 | 3,505.49 | 18,300.83 | 49,184.05 | – |
| CPU (4 threads) | 2,245.61 | 1,023.48 | 5,460.26 | 21,169.81 | 89,196.19 |
| NP | 1,585.69 | 697.51 | 4,407.42 | 6,154.18 | 44,137.50 |
| WE | 1,398.47 | 612.14 | 3,742.69 | 4,796.71 | 37,803.60 |
| NP+warp32 | 511.73 | 196.67 | 1,497.56 | 13,883.50 | 32,567.60 |
| WE+warp32 | **403.51** | **159.68** | 1,214.86 | 4,969.10 | 25,382.20 |
| WE+warp4 | 784.86 | 327.93 | 1,901.29 | 4,593.97 | 28,315.70 |
| WE+warp8 | 562.48 | 229.53 | 1,365.80 | **4,579.23** | 25,469.50 |
| WE+warp16 | 439.58 | 174.51 | **1,170.26** | 4,706.05 | **24,715.40** |
| best speedup (over sequential CPU) | 18.57× | 21.95× | 15.64× | 10.74× | – |
| best speedup (over parallel CPU) | 5.57× | 6.41× | 4.67× | 4.62× | 3.61× |

| Algorithm | rt-higgs | mt-higgs | rec-amazon | sc-shipsec1 | soc-pokec | kron_g500 |
|---|---|---|---|---|---|---|
| CPU (sequential) | 54,717.96 | 1,829.63 | 2,116.82 | 9,763.16 | – | – |
| CPU (4 threads) | 21,522.20 | 746.84 | 656.59 | 4,046.30 | – | – |
| NP | 4,681.37 | 222.60 | 309.49 | 889.72 | – | – |
| WE | 4,197.30 | 197.24 | **226.55** | 776.21 | – | 317,922.00 (88.3 h) |
| NP+warp32 | 6,205.65 | 337.89 | 1,115.83 | 1,478.78 | – | 245,450.82 (68.2 h) |
| WE+warp32 | 4,757.07 | 215.46 | 282.98 | 527.08 | 284,498.82 (79 h) | – |
| WE+warp4 | **3,574.16** | **166.88** | 229.46 | 502.52 | – | – |
| WE+warp8 | 3,641.77 | 169.14 | 238.03 | **479.26** | – | – |
| WE+warp16 | 4,008.30 | 184.45 | 255.84 | 484.18 | – | – |
| best speedup (over sequential CPU) | 15.31× | 10.96× | 9.34× | 20.37× | – | – |
| best speedup (over parallel CPU) | 6.02× | 4.48× | 2.90× | 8.44× | – | – |

Notes:

The times are expressed in seconds. The last two rows report speedups. The result of the CPU sequential algorithm on the actors network cannot be provided because this program consumed too much time on this network. For the same reason, we also ran only selected GPU algorithms on the two very large networks. Bold entries display running times of the fastest algorithms for specific networks.

these networks, as shown in Table 2, and we will also further demonstrate this later. With an appropriate adjustment of _WARP_SIZE_ for low-degree networks, the best-performing program achieves average speedups of 5.2× compared with the parallel CPU implementation and 2.34× compared with the baseline node-parallel strategy. For the rec-amazon graph, which has the lowest maximum degree, the load-imbalance problem does not exist, and for this reason, the warp-centric method cannot improve the performance; instead, the algorithm in which the work-efficient strategy alone is applied performs the best.

### Influence of network structure

To deeply investigate the relationship between network structure and performance for the four GPU implementations, we further ran these algorithms on two types of synthetic graphs, with the results shown in Fig. 3. From Figs. 3A to 3D, we find that the work-efficient algorithm performs better than the node-parallel algorithm on all networks since it always reduces the required number of threads. As seen in Figs. 3A and 3B,

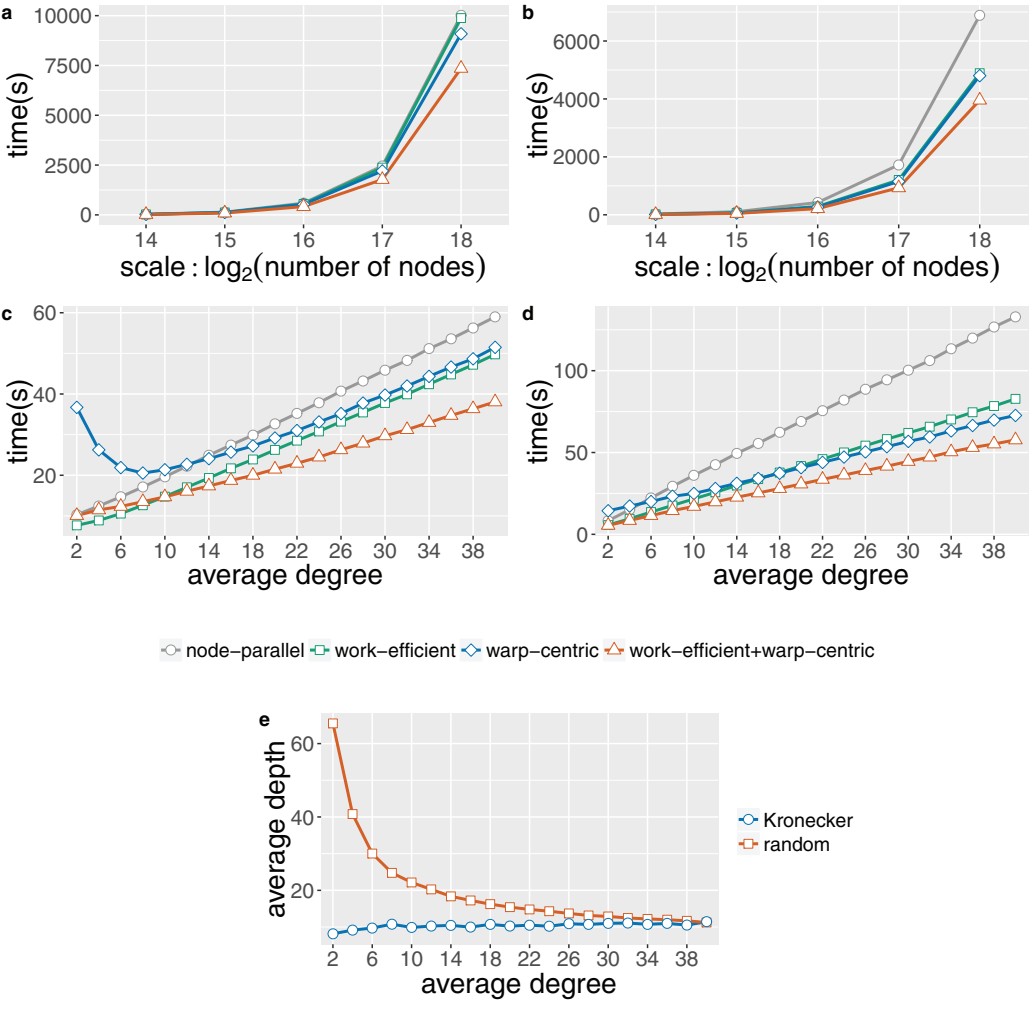

**Figure 3 Performance of the four implementations on ER random and Kronecker graphs.**
Here, *WARP_SIZE* is fixed to 32 in the two warp-centric methods. (A) and (B) show the results of
varying the number of nodes from $2^{14}$ to $2^{18}$ for ER random and Kronecker graphs, respectively, with a
fixed average degree of 32 for both types of networks. (C) and (D) show the results of varying the average
degree for random and Kronecker networks, respectively, where the random networks contain 20,000
vertices and the Kronecker networks contain $2^{15}$ nodes. (E) Illustrates the average depths of the search
trees used for the random graphs in (C) and the Kronecker graphs in (D). Networks with greater depths
have smaller average frontier sets, resulting in poor parallel performance.

the warp-centric method works well on networks of high degrees, which is consistent
with the findings for the real-world networks. Note that for Kronecker graphs, the
warp-centric method works better than it does for random graphs since Kronecker graphs
have a severe load-imbalance problem, which the warp-centric technique can
appropriately address. By contrast, for ER random graphs, as shown in Fig. 3A, the only
advantage of the warp-centric method is its efficient memory access. For low-degree
graphs, the warp-centric method results in even worse performance than the node-parallel
strategy, as can be seen in Figs. 3C and 3D, because the degrees are always smaller than
*WARP_SIZE*. For random graphs, the performance of the warp-centric method is

**Table 3 The results of using different values of *WARP_SIZE* on several low-degree networks.**

| Network | rt-higgs | mt-higgs | ER | Kronecker |
|---|---|---|---|---|
| node-parallel | 4681.37 | 222.60 | 550.86 | 197.54 |
| warp4 | 4182.84 | 205.47 | 578.09 | 159.74 |
| warp8 | 4446.10 | 222.39 | 629.71 | 162.92 |
| warp16 | 5088.25 | 260.28 | 805.60 | 179.60 |
| warp32 | 6205.65 | 337.89 | 1294.71 | 250.16 |
| WE | 4197.30 | 197.24 | 435.93 | 133.84 |
| WE+warp4 | 3574.16 | 166.88 | 413.36 | 118.90 |
| WE+warp8 | 3641.77 | 169.14 | 420.20 | 115.25 |
| WE+warp16 | 4008.30 | 184.45 | 449.54 | 115.12 |
| WE+warp32 | 4757.07 | 215.46 | 501.69 | 118.61 |

**Notes:**
WE is short for work-efficient algorithm. Both the ER network and the Kronecker network have $2^{17}$ nodes, and the average node degree of each is four. The times are expressed in seconds. For these low-degree networks, implementations with smaller *WARP_SIZE* values achieve better performance.

extremely poor when the average degree is smaller than 8, and Fig. 3E illustrates the reason. The low average degree results in a large average depth, which means that the average size of the frontier sets is small. In this case, the warp-centric method assigns more useless threads to nodes that do not need inspection. However, as the degree grows and approaches *WARP_SIZE*, the depth simultaneously drops sharply, which makes the warp-centric method perform much better. Meanwhile, low-degree Kronecker graphs have power-law degree distributions and small average depths; consequently, the warp-centric method does not perform as poorly as on random graphs. However, the combination of these two methods always results in faster performance than the work-efficient method alone because it avoids the second disadvantage of the warp-centric method, as discussed in the previous section. In conclusion, the work-efficient method always achieves better performance, whereas the performance of the warp-centric method depends on the network structure; however, an algorithm that combines the two always achieves the best performance.

### Analysis of warp size

As seen from the above analysis, using a smaller *WARP_SIZE* may accelerate both the node-parallel and work-efficient implementations combined with the warp-centric method when the average degree of the network is small. This hypothesis is verified in Table 3. We applied smaller *WARP_SIZE* values on the rt-higgs network, the mt-higgs network and two synthetic graphs with an average degree of four. We find that implementations with smaller *WARP_SIZE* values perform better than either the baseline node-parallel algorithm or the algorithm with the largest *WARP_SIZE* on both of the low-degree real-world networks, rt-higgs and mt-higgs. Moreover, when coupled with the work-efficient method, algorithms with smaller *WARP_SIZE* values also perform better than either the work-efficient strategy alone or the combination of the work-efficient strategy and the largest *WARP_SIZE*. The reason is that a small *WARP_SIZE* reduces the required number of threads, thereby eliminating the waste incurred when the number

of threads assigned to a node is greater than its degree. The implementations with small *WARP_SIZE* values coupled with the work-efficient method achieve the best performance because they avoid both disadvantages of the warp-centric method while utilizing its advantages. The results obtained on the low-degree Kronecker graph are similar to those obtained on realistic networks for the same reason. For ER random graphs, algorithms with smaller *WARP_SIZE* values do not achieve better performance compared with the node-parallel version because of the large average tree depth, as discussed in the previous section. However, when the work-efficient method is applied, implementations with smaller *WARP_SIZE* values perform slightly better than the work-efficient algorithm alone, thereby further demonstrating the excellent performance and stability of the joint algorithm. In summary, the joint algorithm is the most efficient and the most insensitive to the network structure. Moreover, if we choose an appropriate *WARP_SIZE* for the graph of interest, the performance of the joint algorithm can be even further improved (see Tables 2 and 3).

## CONCLUSION

Existing GPU versions of BC algorithms have concentrated only on unweighted networks for simplicity. Our work offers an algorithm for computing BC in large weighted networks, bridging this gap and enabling a marked efficiency enhancement compared with CPU implementations. Moreover, we incorporate two excellent techniques into our algorithm: the work-efficient and warp-centric methods. The work-efficient method allocates threads more efficiently, and the warp-centric method solves the load-imbalance problem while simultaneously optimizing memory access. We have compared these implementations with sequential and parallel CPU algorithms on realistic networks. The results show that the GPU parallel algorithms perform much better than the CPU algorithms and that the algorithm that combines both the work-efficient and warp-centric techniques is the best, achieving 2.9× to 8.44× speedups over the parallel CPU version and 10× to 20× speedups over the sequential CPU version. Results obtained on synthetic random graphs and Kronecker graphs further demonstrate the superior performance of our solution.

In our future work, we will consider other techniques for addressing the load-imbalance problem to further improve the performance of our algorithm (*Davidson et al., 2014*; *Wang et al., 2016*). In addition, *Solomonik et al. (2017)* have proposed a parallel BC algorithm for weighted graphs based on novel sparse matrix multiplication routines that has achieved impressive performance, which may provide further inspiration for accelerating our algorithm. We may also consider implementing a GPU algorithm for processing dynamic networks. When networks change only slightly (e.g., a few new nodes are added or a few links vanish), recalculating the BC for all nodes is unnecessary because the BC of most nodes and edges will not change. Several previous works have explored sequential algorithms for addressing this issue (*Lee, Choi & Chung, 2016*; *Singh et al., 2015*; *Nasre, Pontecorvi & Ramachandran, 2014*). We plan to develop a GPU version of these algorithms to achieve better performance.

### Funding

This work was supported by NSFC (Grant No. 71501005) and the fund of the State Key Laboratory of Software Development Environment (Grant Nos. SKLSDE-2015ZX-05 and SKLSDE-2015ZX-28). Rui Fan is supported by the Innovation Foundation of BUAA for PhD Graduates. The funders had no role in study design, data collection and analysis, decision to publish, or preparation of the manuscript.

### Grant Disclosures

The following grant information was disclosed by the authors:
NSFC: 71501005.
State Key Laboratory of Software Development Environment: SKLSDE-2015ZX-05 and SKLSDE-2015ZX-28.
Innovation Foundation of BUAA for PhD Graduates.

### Competing Interests

The authors declare that they have no competing interests.

### Author Contributions

- Rui Fan conceived and designed the experiments, performed the experiments, analyzed the data, wrote the paper, prepared figures and/or tables, performed the computation work.
- Ke Xu conceived and designed the experiments, wrote the paper, reviewed drafts of the paper.
- Jichang Zhao conceived and designed the experiments, wrote the paper, reviewed drafts of the paper.

### Data Availability

Zhao, Jichang (2017): A GPU-Based Solution to Fast Calculation of Betweenness Centrality on Large Weighted Networks. figshare.

https://doi.org/10.6084/m9.figshare.4542405.v2

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
