# Peer review of "A GPU-based solution for fast calculation of the betweenness centrality in large weighted networks"

_PeerJ Computer Science, doi:10.7717/peerj-cs.140_

## Round 0.1 · original submission · Major Revisions

Thank you for submitting your work to PeerJ. As editor, I believe the three reviews were fair and constructive, and I agree with, and recommend, their consensus recommendation of "major revisions". I believe that implementing their suggestions will both significantly improve the work and its impact as well as lead to acceptance in PeerJ.

The reviewers all agree that the topic of the work is interesting and important, and that the topic has not previously been addressed in the literature. (I agree!) In my opinion the suggestions from the reviewers are in every way appropriate and helpful, and in their response, I expect the authors to address every one of them, even if it is to explain why they did not accept or address the suggestion. I believe the most important aspects to address are:

- Concern about the breadth of datasets and the suitability of ER and Kronecker synthetic graphs. Please also add a road-network-like graph (which will have a much smaller max degree)
- Concern about the size of graphs; considerably larger graphs are the standard in GPU graph analytics today
- Improve the breadth of and detailed comparison to previous work
- Better structure for results section
- Address Reviewer 2's question about locks
- Improvement of text quality; all three reviewers noted that the English had significant room for improvement

I note that the abstract of this work is currently more like an introduction. It would benefit from an edit that made it more of an abstract, which is a concise summary for experts; note https://www.lightbluetouchpaper.org/2007/03/14/how-not-to-write-an-abstract/

I also expect that in the bibliography, relevant words are capitalized ("GPU", "CUDA", etc.), which you can do by putting them in braces in your bibtex (e.g., "Accelerating the {XGB}oost algorithm using {GPU} computing"). All cited work needs full citations, including volume/issue/page information for journals, pages for conferences, proper capitalization for venue titles, and URLs for online work that is not available anywhere else. Please fix Erdos's coauthor.

·

Basic reporting

* In many places, authors cite some work without explaining what’s actually done in that work -- it'd be quite hard for a reader without a background on BC. For example, line 221 says “In this paper, we apply the novel warp-centric method (Hong et al., 2011), which allocates a warp rather than a thread to one node”. Could you please elaborate on that? Another example is line 208: “McLaughlin et al. figured out an
excellent work-efficient technique to solve this problem (McLaughlin and Bader, 2014)”. The following few sentences are not enough to present that work. I strongly recommend having a separate “related work” section before the “GPU-based algorithm” section that summarizes those previous works in detail. For each work that you summarize, please stress the differences and similarities of your work.

* Pseuducodes should help understand the algorithm. No need to give every single detail (like Alg. 1) to show the mechanics. Please shorten those. Using a different tex package that does not show “end if” and “end for”s would also be great (indentations do the same thing).

* Algorithm section basically tells the mechanics of the proposed solution without any high-level explanation or intuition. Please provide a high-level explanation of your algorithm first, then give the details.

* Background is not detailed enough, please give the pseudocode for Dijkstra’s alg.

* ln 149: explain coarse-grain parallelization, which is also used in exps (ln 245).

* So many grammar mistakes (esp articles), some are below. Multiple proofreadings needed.
- ln 23: publicly
- ln 38: While Brandes .. -> Brandes ..
- ln 44: millions of active
- ln 68: characteristics
- ln 93: Brandes
- ln 99: how to adapt for directed?
- ln 102: the minimum length of all paths connecting s and t -> the length of the shortest path connecting s and t
- ln 104: an undirected graph
- ln 113: for an unweighted .. for a weighted ..
- ln 118: an unweighted
- ln 120: tree is
- ln 126: the shortest path problems. Dijsktra algorithm
- ln 129: s to an

* Please format the last two lines of Tab. 2 to highlight that they're speedups (not runtimes).

* Replacing Fig 4 by a table (like Tab. 2) might be better. The current indirection for the algorithm variants (A, B, C ..) makes it hard to understand.

Experimental design

Regarding the motivation:

* ln 17: Not sure if the weighted networks are more prevalent than the unweighted ones. Yes, there exist attributes on the edges in many real-world data, but capturing those with the weights is a hard problem and not always possible. Please reword that.

* ln 49: It cannot be said that GPGPU computing provides excellent parallelization for graph/network algorithms. It is always nontrivial to get good performances for the irregular computations (that's why this work is interesting).

For the experiments:

* ER and Kronecker graphs are not realistic models and do not have the community structure as in the real-world data, please check "Community Structure and Scale-free Collections of Erdős-Rényi Graphs" by Seshadhri et al. In addition, uniformly assigning random edge weights is not justified -- being used by previous works cannot be a reason.

* The largest network has ~15M edges, that's not big enough to test the possible memory constraints of GPU.

* I wonder if the results would be different for the networks that have integer weights and double weights. No information is given for the used data. I suspect double weighted networks might be cheaper for BC computation since there would be less number of shortest paths between any pair of nodes.

Validity of the findings

- ln 99: how to adapt for directed? Please give more detail or don't say that.

- There is a contribution in this work, but the presentation must be improved significantly and a few issues about the exp design should be addressed.

Additional comments

Ignoring the presentation and the data used in exps, I believe that authors provide a nice combination of the existing techniques (for unweighted BC) and adapt them for the weighted BC computation.

·

Basic reporting

This manuscript has many minor grammatical issues and would greatly benefit from a review by a native English speaking colleague. Here are some examples:

- The abstract contains the phrase "can arrive the performance of." This phrase should be replaced with "attains a" or "achieves a" when mentioning the speedup of the algorithm.
- The use of the word "while" on line 38 is confusing.
- Line 50: I would prefer the term "applications" (or a synonym of it) to "issues"
- Line 52: I would rephrase this sentence as "CUDA, developed by NVIDIA, is the most popular..."
- Line 130: "According to be settled or not" could be rephrased as "According to whether or not a node is settled"

The reference supplied by this article are sufficient, although I would add a reference to the following paper:

E. Solomonik et al., "Scaling betweenness centrality using communication-efficient sparse matrix multiplication."

Also, on Line 75 there is a citation to "Merrill et al., 2015" which does not match any of the references. Perhaps you meant the paper from Jia et al. in 2011?

Experimental design

The research question of this work is well-defined. One particular aspect of the work I would like to see a clearer explanation of is the use of a lock. Locks are particularly problematic in CUDA as they're not only error-prone but also can halt entire warps instead of individual threads as they would on a CPU. Interestingly, CUDA 9.0's capabilities might have a big influence of this choice of algorithm.

In fact, CUDA 9.0 may also come into play when attempting to figure out the appropriate value of WARP_SIZE, as CUDA 9.0 supports cooperative groups. Since you used CUDA 8.0 (which is obviously fine), do you see a way that you could determine the best value of WARP_SIZE dynamically without having a significant performance impact?

Validity of the findings

The depth of experiments and high quality of results are the strongest point of this submission. The conclusions drawn by the authors support those of prior work and appear reasonable.

·

Basic reporting

The paper describes an efficient GPU algorithm for computing betweenness centrality on large-scale weighted graph, and shows performance advantage via extensive experiments and performance analysis. Overall, the idea is clearly presented with sufficient literature references and background/context. That said, the English expression of a lot of the details of algorithm description and performance analysis needs to be improved. Most figures and tables are self-explanatory, one general suggestion is that the author should consider putting relevant figure/table at the same page as the texts.

Experimental design

The research question is well-defined and the open-sourced implementation will greatly help the community to reproduce and improve on this work.

On the experimental design, I have two high-level comments:
1) Comparing to other betweenness centrality papers, especially [McLaughlin 2014], the datasets in this paper is not large enough. I suggest the author to include at least one graph with more than 1 million nodes for each synthesis dataset and real-world dataset.

2) The current results analysis is not organized well enough. Could the authors restructure the section into something similar to "overall performance", "analysis on different traversal strategies", "analysis on different datasets"? (Not necessarily the one I suggested, but just structure the section in this way will help audience understand the experiments better)

3) For the sequential and multi-thread CPU implementations, are they the state-of-the-art implementation? Please add more description and/or citation information of these two algorithms.

Also, on page 10/14:308, the authors claim that R-Mat and real-life scale-free graphs have similar topology. I have different opinion on that, since R-Mat graphs are more scale-free than real-world graphs, real-world scale-free graphs usually have lower edge factor. The authors should consider adding analysis according to different edge factors.

Validity of the findings

The main problem is that the paper's method is not innovative enough as both the graph traversal part and workload balancing strategy part is based on previous methods. But I think the topic is worth exploring, since it is true that no work has been done on weighted graph betweenness centrality on the GPU.

The main things this paper could improve are as follows:

1) It does not try another method which simultaneously does graph traversal and can potentially speedup betweenness centrality: 10.1109/ICPADS.2015.10
If the authors think there is technical reason this method cannot be applied, they should include this in the literature review and specify the reason it is not applied in this paper.

2) For the load-balancing strategy, the authors tried Merrill's method (referred to as work-efficient method) and warp-centric method, but they miss the other strategy which were used in both [Davidson 2014] and [Wang et al. 2016]'s work. The merge-path based load-balanced search has better performance on scale-free graphs on all traversal-based graph algorithms, which I believe includes betweenness centrality too.

3) One minor pick is that I think with small modification, delta-stepping could also level information and number of shortest paths information. In that case, using delta-stepping with a smaller number of buckets could also potentially increase the performance. The authors should provide more details on the reason of not using delta-stepping method.

Additional comments

I put several grammar comments and minor technical comments in the attached pdf file.

---

## Round 0.2 · accepted · Accept

Thank you for carefully and thoughtfully considering and replying to the concerns of the reviewers. I am happy to recommend acceptance of this manuscript.

Because I care too much about typography, may I make the following typographic suggestions which can be addressed while in production:

- \usepackage{cite} to alphabetize citations
- use \shortcite (if the package allows it) to avoid repetition of author names: e.g., "work of Brandes (Brandes, 2001)" should be "work of Brandes~\shortcite{Brandes:2001}", which will typeset as "work of Brandes (2001)". Also "However, as described in (Davidson et al., 2014)" should be "However, as described in Davidson et al.~\shortcite{Davidson:2014}"
- Make sure that all "Author et al. did ..." is typeset as "Author et al.\ did ..." (with inter-word space not inter-sentence space, use the backslash)
- Use \textit{} within math mode to typeset words in math mode. Don't just use plain text inside $$. https://tex.stackexchange.com/questions/345373/is-the-effect-of-dollar-sign-the-same-as-textit
- Consider using the booktabs package to typeset your tables. Its intro material talks a lot about how to typeset good tables.

Bibliography:

- Harish needs a venue. I don't think you're using the right bibtex cite type.
- Use {} in titles {CUDA}, {SSSP} for the Martin cite and {BGL} (Gregor).
- De Domenico and Gregor need more complete citations. Gregor: fix the venue.
- Nasre: What is the venue where this was published?
- Rossi: Pages?
- Don't use {{}} to wrap titles (Solomonik). Use sentence case for them, not title case.
- Collapse the two Wang cites into one (presumably the later one).

·

Basic reporting

No comment.

Experimental design

No comment.

Validity of the findings

No comment.

Additional comments

I'm satisfied with the revision. Organization and presentation are well done, and the paper is comprehensive now for a general CS audience.

·

Basic reporting

The revised manuscript is better structured and has greatly benefited from the review of native English speaker(s). The authors took care to cite additional references and provide additional background based on the first set of reviews.

Experimental design

no comment

Validity of the findings

no comment

Additional comments

Overall I thought the authors did a fantastic job of restructuring and cleaning up this manuscript. It now provides a thorough understanding of the complications of computing between centrality for weighted networks, with a nice history of recent work in this area. I feel that this manuscript is now ready for publication.

·

Basic reporting

The author has made significant improvement on writing and the structure of the paper. It added more citations, explanations to previous unclear descriptions, as well as more experiments and data sets.

Experimental design

Previously my main concerns were: 1) not comparing with the SOTA GPU weighted-BC algorithm; and 2) not having large graph data sets.
In the revised version, the author proved that there is no weighted-BC implementation on the GPU, and it is difficult to modify any unweighted-BC implementation. Instead, work in this paper could serve as a good baseline weighted-BC algorithm.
The revised paper also included two large graphs (one synthesized, one real). However, no CPU performance is presented, which is strange since usually size of graph should not be a limiting factor for CPU algorithms. Author should add some explanation to this.

In general, I really like the re-structured experiment section.

Validity of the findings

I think with the revision and the open-sourced implementation, the paper could serve as a baseline for weighted-BC algorithm on the GPU.

Additional comments

Thanks for the revision and carefully language improvements. The revised version looks much better.